# Gre Factors Are Required for Biofilm Formation in *Salmonella enterica* Serovar Typhimurium by Targeting Transcription of the *csgD* Gene

**DOI:** 10.3390/microorganisms10101921

**Published:** 2022-09-27

**Authors:** Tania Gaviria-Cantin, Andrés Felipe Vargas, Youssef El Mouali, Carlos Jonay Jiménez, Annika Cimdins-Ahne, Cristina Madrid, Ute Römling, Carlos Balsalobre

**Affiliations:** 1Department of Genetics, Microbiology and Statistics, School of Biology, University of Barcelona, Av. Diagonal 643, 08028 Barcelona, Spain; 2Department of Microbiology, Tumor and Cell Biology, Karolinska Institutet, 17177 Stockholm, Sweden

**Keywords:** biofilm, Gre factors, *Salmonella*, CsgD expression, transcriptional pausing

## Abstract

Rdar biofilm formation of *Salmonella typhimurium* and *Escherichia coli* is a common ancient multicellular behavior relevant in cell–cell and inter-organism interactions equally, as in interaction with biotic and abiotic surfaces. With the expression of the characteristic extracellular matrix components amyloid curli fimbriae and the exopolysaccharide cellulose, the central hub for the delicate regulation of rdar morphotype expression is the orphan transcriptional regulator CsgD. Gre factors are ubiquitously interacting with RNA polymerase to selectively overcome transcriptional pausing. In this work, we found that GreA/GreB are required for expression of the *csgD* operon and consequently the rdar morphotype. The ability of the Gre factors to suppress transcriptional pausing and the 147 bp 5′-UTR of *csgD* are required for the stimulatory effect of the Gre factors on *csgD* expression. These novel mechanism(s) of regulation for the *csgD* operon might be relevant under specific stress conditions.

## 1. Introduction

The ubiquitous transcription elongation factors GreA and GreB, henceforth jointly named Gre factors, small cytoplasmic proteins of <20 kDa, have been extensively studied in *Escherichia coli*. Several roles have been attributed to those proteins: (i) stimulating transcription initiation by likely facilitation of promoter escape, (ii) promoting release of stalling RNA polymerases during transcription elongation by inducing the intrinsic endoribonucleolytic activity of the RNA polymerases, and (iii) chaperone activity promoting refolding of denatured proteins [1,2,3]. In *E. coli* we have previously described that the Gre factors affect expression of biofilm promoting factors such as type 1 fimbriae and flagella [4,5]. Moreover, transcriptome studies using *E. coli* K-12-derivative strains allowed the conclusion that genes upregulated by GreA are mostly involved in energy metabolism and protein synthesis, whereas most of the genes downregulated by GreA belong to the cell metabolism and stress response [6]. Of note, a more recent report systematically characterized the promoter proximal pauses occurring during transcription in *E. coli*. It has been suggested that transcriptional pausing represents a widespread regulatory mechanism aimed at modulation of transcriptional output from specific operons in response to changing environmental conditions in a Gre factor dependent manner [3].

In *Salmonella enterica* serovar Typhimurium (*S.* Typhimurium), while explored to a lesser extent, it has been shown that the Gre factors play a major role in *S.* Typhimurium pathogenic lifestyle. We showed that a strain lacking the Gre factors was avirulent both in vitro and in vivo. Specifically, the expression of the SPI-1 genes, mediating host cell invasion, was importantly down regulated in the absence of the Gre factors due to a direct effect on the transcription elongation of the *hilD* gene, coding for the master regulator of the SPI-1 genes. A transcriptional pause in the 3′-UTR of the *hilD* gene was the target for the Gre factors [7]. *S.* Typhimurium has the ability to form biofilm on diverse biotic and abiotic surfaces, such as interacting with epithelial cells during the infection process [8] and adhering to abiotic surfaces such as plastic, glass and steel, leading to transmission and outbreaks of *S.* Typhimurium during food processing [9]. Most studies on *S.* Typhimurium biofilm have been performed upon growth on/in rich media, low osmolarity and low temperature (25–28 °C). Under these conditions, *S.* Typhimurium produces a rdar colony morphotype on agar plates and a pellicle air-liquid interface biofilm [10]. Two pivotal interaction-promoting factors are the main components of the extracellular matrix of described biofilms: curli, an amyloid proteinaceous surface appendage, and cellulose, an exopolysaccharide. Curli production is encoded in the divergently transcribed biogenesis operons *csgBAC* and *csgDEFG*. The biosynthesis of cellulose requires proteins encoded by the divergently transcribed operons *bcsQRABZC* and *bcsEFG*. Production of c-di-GMP by AdrA, a diguanylate cyclase encoded in the *adrA* gene, is necessary to eventually post-transcriptionally activate biosynthesis of cellulose. CsgD, an orphan transcriptional regulator encoded in the *csgDEFG* operon, regulates production of both curli and cellulose by stimulating transcription of the *csgBAC* and *adrA* genes [11,12]. CsgD expression is subject to complex regulation at the transcriptional level mediated by a 521 bp long intergenic region in combination with sequences downstream of the translational start site [13,14]. A 147 bp long 5′-UTR required for optimal expression is targeted by small regulatory RNA regulation [15,16,17,18]. The genes and basic regulation of rdar morphotype expression are conserved between *S.* Typhimurium and *E. coli* [19].

The putative widespread role of Gre factors on transcription elongation and its previously described key role in fundamental regulatory pathways for both *E. coli* and *S.* Typhimurium indicated that Gre factors might play a role in the regulation of the ability of *S.* Typhimurium to form biofilms. Here, the role of the Gre factors in regulating the process of biofilm formation in *S.* Typhimurium has been explored. We found that biofilm production was impaired in strains lacking the Gre factors. Consistently, the production of curli and cellulose was found to be strictly dependent of the Gre factors being abolished in strains lacking both factors. Expression studies showed that the Gre factors are required for the transcriptional expression of the master regulator CsgD. Gre-mediated regulation of *csgD* does not occur at transcription initiation but on the level of transcription elongation within the 5′-UTR of *csgD*. Our data add new players to the complex regulation of the biofilm formation process in *S.* Typhimurium and highlight the impact of 5′-UTRs of mRNAs as key regulatory hubs, not only at the posttranscriptional level but also at the level of Gre-mediated transcriptional elongation.

## 2. Materials and Methods

### 2.1. The Bacterial Strains and Culture Media

The *Salmonella enterica* serovar Typhimurium (*S*. Typhimurium) wild-type strains used in this work are UMR1, a rdar_28_^+^ nalidixic acid resistant derivative of ATCC14028 [20], and SV5015, a His^+^ derivate of SL1344 [21,22]. All strains and plasmids used in this work are described in Appendix A. Cultures were routinely grown on either LB (10 g/L NaCl, 10 g/L tryptone, 5 g/L yeast extract and 15 g/L agar), LB_0_ (10 g/L tryptone, 5 g/L yeast extract and 15 g/L agar) or CFA plates (10 g/L casamino acids, 1.5 g/L yeast extract, 0.4 mM MgSO_4_ and 0.4 mM MnCl_2_). When required, ampicillin (50 μg/mL), kanamycin (50 μg/mL), chloramphenicol (25 μg/mL), spectinomycin (25 μg/mL), arabinose (0.02%) and calcofluor (Fluorescent Brightener 28, Sigma-Aldrich, Saint Louis, MO, USA) (0.2%) were added.

### 2.2. Genetic Manipulations

Chromosomal *lacZ* fusions were generated in two subsequent steps. First, homologous recombination by one-step gene replacement [23] was applied to partially delete the *csgD* gene from different positions (+9 and +147). Subsequently, a promoter-less *lacZ* gene was integrated by site-specific recombination via a suicide plasmid [24] (see Appendix A. Briefly, hybrid primers containing nucleotide sequences homologous to the 5′ (csgD+9 and csgD+147) and 3′-end (csgD lac Rev) and sequences flanking the antibiotic marker were used to amplify the kanamycin resistance cassette encoded by plasmid pKD4. Gene replacement and removal of the antibiotic cassette in *S*. Typhimurium SV5015 was performed by the standard approach [23]. Constructs were PCR-verified with primers csgD up/csgD down located in the sequences flanking the deleted open reading frame (Appendix A). The suicide plasmid pKG136, bearing a FRT site upstream of a promoter less *lacZ* gene, was electroporated into the resulting strains transformed with plasmid pCP20. The pKG136 plasmid was integrated into the chromosome by FLP-mediated recombination aided by plasmid pCP20. Recombinant clones were selected on LB agar containing kanamycin and X-Gal. Primers used in this work are listed in Appendix A.

Mutant alleles were transduced into a novel strain background by phage P22 HT/int4 [25]. Transductants were streaked twice on EBU LB-agar plates (0.25% (*w*/*v*) glucose, 0.25% (*w*/*v*) K_2_HPO_4_, 0.0125 g/L Evans blue (Sigma-Aldrich, Saint Louis, MO, USA) and 0.0250 g/L fluorescein (Sigma-Aldrich, Saint Louis, MO, USA)) [26] supplemented with the corresponding antibiotics.

The 5′UTR of *csgD* was PCR amplified using primers csgD UTR Fw/csgD UTR Rv carrying restriction sites for NcoI and SalI, respectively. The amplified fragment was cloned in pGEM-T easy (Promega corporation, Madison, WI, USA). The cloned fragment was sequenced and subcloned into the NcoI and SalI restriction sites of the pTT68 plasmid.

### 2.3. Biofilm Formation, Detection and Quantification

For biofilm on glass, qualitative assays were performed as described [27]. For quantitative assays on plastic, CFA cultures were grown using flat-bottomed 96-well polystyrene plates (Nalge Nunc International, Roskilde, Denmark). The inoculum was achieved by diluting a *S.* Typhimurium suspension to an OD_600nm_ of 0.02. Plates were incubated for 72 h at 28 °C in static conditions inside a plastic bag with wet cellulose paper to maintain the humidity levels. Liquid culture was removed and the OD_600nm_ determined. Quantitative measurement of the biofilm biomass was performed by crystal violet (CV) staining as described [28], with minor modifications. Briefly, plates were rinsed twice with distilled water and biofilms were fixed by heating at 80 °C for 30 min. Then, 0.2 mL of a 1% (*w*/*v*) CV solution (Panreac Applichem, Darmstadt, Germany), was added to the wells and incubated for 15 min, rinsed with water and air-dried. 0.2 mL of acetic acid 30% (*v*/*v*) was added to the wells. The OD_570nm_ of the resulting solution was determined. The signal detected in control wells containing only culture media was subtracted from the values obtained for all the samples. The data are mean values with standard deviations of six independent cultures.

### 2.4. β-Galactosidase Assay

Cultures were grown on LB_0_ plates at 28 °C for 24 or 48 h. After incubation, bacterial suspensions in LB were obtained, its OD_600nm_ determined and kept on ice until β-galactosidase assay was performed, as described by Miller [29]. Enzymatic activity was determined from three independent cultures with duplicate values and the mean values with standard deviations were plotted.

### 2.5. Cellulose Quantification

Qualitative assessment of bacterial cellulose production was determined on LB_0_ plates supplemented with calcofluor. Plates were incubated at 28 °C for 48 h and fluorescence was observed under a UV light source. 

### 2.6. Rdar Morphotype 

The rdar morphotype was judged visually on Congo red agar plates (LB_0_ agar plates supplemented with Congo red (Fluka Chemie GmbH, Buchs, Switzerland) (40 mg/L) and Coomassie brilliant blue G-250 (Sigma-Aldrich, Saint Louis, MO, USA) (20 mg/L)). Five microliters of a bacterial suspension in water (OD_600nm_ of 5.0) from a fresh LB-agar plate were spotted onto the plate and incubated at 28 °C.

### 2.7. Immunodetection of CsgD

Immunodetection of CsgD was performed as described [7], using a polyclonal peptide antiserum against CsgD [12].

### 2.8. Expression Analysis by qPCR

RNA was isolated from cultures grown on LB_0_ plates at 28 °C for 24 h, as described previously [30]. qPCR assays were performed as described [7]. The primer pair used for *csgD* transcript detection was qCsgDfw/qCsgDrev. Expression was normalized against the *recA* gene as endogenous control using the primer pair qrecAFW/qrecAREV. 

### 2.9. Statistical Analysis

Differences between average values were tested for significance by performing an unpaired two-sided Student’s test. The levels of significance of the resulting *p* values are indicated in the figure legends.

## 3. Results and Discussion

### 3.1. Rdar Biofilm Formation Is Impaired in S. Typhimurium When Deficient for the Transcription Elongation Factors GreA and GreB

*S.* Typhimurium is able to form a *csgD*-dependent rdar biofilm on a plethora of surfaces. In order to address the potential requirement of the Gre factors for the ability of *S.* Typhimurium to form biofilm, we determined the capacity of *Salmonella enterica* SL1344 *his*^+^ derivative SV5015 (Wt) strain and its mutants ∆*greA*, ∆*greB* and ∆*greA*∆*greB* (from now referred as ∆*greAB*) to form biofilm on glass and plastic. Both the qualitative biofilm assay on glass and quantitative determination of biofilm formation on plastic indicated that biofilm production was impaired in the absence of both Gre factors (Figure 1A). The absence of either *greA* or *greB* did not abolish biofilm formation, although surface-growth of cells was reduced to some extent with a more pronounced drop in biofilm biomass for ∆*greA* than for ∆*greB*. The percentage of biofilm biomass compared to Wt was 42% and 87% for ∆*greA* and ∆*greB*, respectively. To corroborate the involvement of the Gre factors in the ability of *S.* Typhimurium to form biofilm, similar experiments were performed with the strain UMR1, a rdar_28_^+^ single colony derivative of ATCC14028 resistant to nalidixic acid, used as a model in biofilm studies [20]. A similar decrease in the ability to form biofilm on polystyrene was observed in the ∆*greA*, ∆*greB* and ∆*greAB* mutant derivatives of UMR1 as described for the SV5015 strains. Consistent with previous reports, the ATCC14028 derivative forms a more pronounced biofilm compared with SV5015, whereas no biofilm was detected from cultures of the ∆*ompR* and ∆*csgD* strains [20] (Figure 1B). 

The production of the two extracellular appendages, curli and cellulose, which are promoting cell–cell, cell–interface and cell–surface interactions in *S*. Typhimurium biofilms, can be phenotypically scored by growing bacteria on Congo red agar plates [31] (Figure 1A,C). As expected, Wt strains, both SV5015 and UMR1, showed a rdar (red, dry and rough) colony morphotype, indicative of curli and cellulose production. Consistent with the qualitative and quantitative biofilm assays, the single mutants—∆*greA* and ∆*greB*—also showed a rdar morphotype, whereas the ∆*greAB* mutant strains showed saw (smooth and white) morphotype, indicating that they do not produce curli and cellulose. The requirement of the Gre factors for the production of the biofilm-promoting factors was further characterized. The transcriptional expression of *csgB*, coding the major subunit of curli, was monitored using a chromosomal *csgB::lacZ* fusion (Figure 1D). Downregulation of *csgB* expression was detected to some extent in the single mutants, being more pronounced in the ∆*greA* mutant. In the absence of both Gre factors, the expression was apparently fully abolished, reaching levels similar to the ∆*ompR* mutant strain. The production of cellulose was determined by growing the different strains in the presence of calcofluor, a fluorochrome that binds to β-1,4-polysaccharides, including cellulose (Figure 1E). Visual examination of the strains growing on LB_0_ supplemented with calcofluor under UV light showed that the Wt strain emitted substantially more fluorescence than the ∆*greAB* mutant, clearly indicating that the double mutant is impaired in cellulose production. Consistently, the single mutants, ∆*greA* and ∆*greB*, showed intermediate calcofluor binding phenotype. As a control, no fluorescence was detected in the ∆*ompR* strain. High emission of fluorescence was observed with the strain MAE52, which semi-constitutively overexpresses *csgD* and the biofilm extracellular matrix. 

Notably, the Gre-mediated control of curli and cellulose expression was further confirmed by successful restoration of the rdar morphotype in a ∆*greAB* strain by trans-complementation with expression of either *greA* or *greB* from a plasmid (Figure 1C). In conclusion, GreA and GreB additively stimulates expression of cellulose and curli in *S*. Typhimurium, resulting in the production of a rdar biofilm. 

### 3.2. The Expression of CsgD, the Master Regulator of Biofilm Formation in S. Typhimurium Is Impaired in Strains Lacking the Gre Factors

In *S.* Typhimurium, the expression of curli and cellulose is under the control of CsgD. CsgD binds to the *csgBAC* promoter, stimulating curli expression, and to the *adrA* promoter, encoding a diguanylate cyclase that promotes cellulose production from the BCS proteins post-transcriptionally through the synthesis of cyclic di-GMP [12]. Since the production of both, curli and cellulose, was impaired in the ∆*greAB* mutant, a feasible explanation is that the Gre factors are required for the expression of CsgD and in turn for the production of the two biofilm-promoting factors. The transcriptional expression of *csgD* was studied by qPCR determination of the *csgD* mRNA levels (Figure 2A). Consistent with previous reports, *csgD* expression was not detected in a ∆*ompR* strain [11]. In agreement with the effect described on the rdar biofilm promoting factors, *csgD* expression was impaired in the ∆*greAB*, whereas the single mutants display partial reduction of *csgD* expression. Furthermore, the production of CsgD was determined by immunodetection of the CsgD protein in whole cell extracts of strains grown on LB_0_ agar plates (Figure 2B). No CsgD was produced when the bacterial cells lack both Gre factors while partial reduction was observed in the absence of ∆*greA* and to a lesser extent on ∆*greB*. Of note, both *greA* and *greB* can complement the drop in CsgD production in a ∆*greAB* strain. Altogether, our data indicated that the Gre factors are required for the expression of *csgD*, the master regulator of rdar biofilm formation in *S.* Typhimurium. Gre factors, as well as the curli operon, are conserved between *S.* Typhimurium and *E. coli*. Not unexpectedly, the absence of the Gre factors caused a severe drop in biofilm production in liquid culture and in the expression of biofilm promoting factors, as depicted by the rdar morphotypes, also in *E. coli* (Figure 3). In summary, *greA* and *greB* upregulate the central hub of the rdar biofilm, *csgD*, on the steady state mRNA level in *S*. Typhimurium and presumably also in *E. coli*. 

### 3.3. The 5′-UTR of the csgD Gene Is Required for Stimulation of csgD Expression by the Gre Factors

In order to determine if the Gre factors are required for *csgD* expression at the level of transcription initiation, transcriptional studies in Wt and ∆*greAB* strains were performed with a chromosomal *lacZ* fusions to the *csgD* gene at position +9 (*csgD*_+9_*::lacZ*, Figure 4A, see materials and methods for details). The down regulation of the *csgD* transcript in the ∆*greAB* mutant, detected by qPCR quantification, was not observed when *csgD* expression was monitored using the *csgD*_+9_*::lacZ* construct (Figure 4B). The *csgD* mRNA carries a long 5′-UTR of 147 bp which is required for enhanced transcription [17] and it is absent in the *csgD*_+9_*::lacZ* construct. In order to determine if the 5′-UTR region is important for the stimulatory role of the Gre factors in *csgD* expression, a *csgD::lacZ* construct carrying the entire 5′-UTR (*csgD*_+147_*::lacZ*, Figure 4A, see materials and methods for details) was generated. When transcription of the *csgD*_+147_*::lacZ* was assessed, a substantial downregulation of *lacZ* activity in the absence of the Gre factors was detected indicating that the 5′-UTR of *csgD* is subject to regulation by *greA*/*greB* (Figure 4B). Thus, we can speculate that Gre factors relieve transcriptional pause(s) in the 5′-untranslated region (5′-UTR) of the *csgD* gene, a mechanism of action occurring in most cases in 5′-UTR regions as described in the literature [3]. 

These results also clearly indicate that the 5′-UTR is required for Gre-mediated stimulation of *csgD* expression, while regulation does not occur at the level of transcription initiation. The latest statement was further corroborated using strain MAE52 carrying a single point mutation in the *csgD* promoter and causing an approximately three-fold higher transcriptional expression of the *csgD* gene. In MAE52, *csgD* is transcribed temperature independently with the aid of the housekeeping sigma factor RpoD instead of the stress sigma factor RpoS as in strain UMR1 [11,32]. Independently of the fact that the *csgD* gene was expressed from a semi-constitutive promoter in MAE52, mutations in *greA* and *greB* lead to downregulation of the expression of the rdar morphotype, which further supports a regulatory mechanism of action by Gre factors downstream of transcription initiation (Figure 4C). 

These data indicate that the 5′-UTR of the *csgD* gene is the target of the Gre factors. The *csgD* 5′-UTR was cloned into the pTT68 plasmid, downstream of a pBAD promoter and upstream of a promoter less *lacZ* gene. Expression of the construct was induced by the addition of arabinose and monitored in both Wt and ∆*greAB* genetic backgrounds. The expression dropped in the ∆*greAB* mutant with the construct carrying the *csgD* 5′-UTR (Figure 5A), further indicating that the 5′-UTR is the target of the Gre factors. 

Gre factors promote transcription elongation by resolving backtracking of paused transcription complexes [1]. In *E. coli*, the D41 and E44 GreA residues are essential for the ability of GreA to alleviate backtracked complexes and thereby to suppress transcriptional pauses [33]. *E. coli* and *S.* Typhimurium GreA share a high percentage of identity (97%) comprising the 41 and 44 residues (Appendix A). Previously, we demonstrated that *E. coli* GreA complements the ∆*greAB* mutant of *S.* Typhimurium and that the D41 and E44 residues are required to suppress transcriptional pausing [7]. To determine whether downregulation of *csgD* expression in the Gre-factor deficient strain is due to a transcriptional pause, two GreA variants of *E. coli* were used, the GreA_Wt_ and the GreA_D41AE44Y_, being the latter impaired to resolve paused transcription complexes. The effect of the presence of GreA_Wt_ and GreA_D41AE44Y_ on the expression of the *csgD* gene was monitored using the *csgD*_+9_::*lacZ* and *csgD*_+147_*::lacZ* constructs (Figure 5B). As expected referring to the previous results, the production of GreA_Wt_ in a ∆*greAB* mutant complemented the absence of the Gre factors as monitored by transcription of the *csgD* fusion carrying the 5′-UTR (*csgD*_+147_*::lacZ*), whereas no change in expression of the *csgD* fusion lacking the 5′-UTR (*csgD*_+9_*::lacZ*). Remarkably, no induction of the *csgD*_+147_*::lacZ* construct was detected when GreA_D41AE44Y_ was expressed, indicating that the reported effect of the Gre factors on *csgD* expression can be due to the resolution of a transcriptional pause occurring in the 5′-UTR of the *csgD* gene. Although, in vitro transcription experiments will be required to confirm the direct involvement of the Gre factors in *csgD* transcription, our data suggest that GreA and GreB alleviate transcriptional arrest in the 5′-UTR of *csgD* mediated by the residues participating in transcript cleavage by the RNA polymerase. Whether additional GreA and GreB checkpoints exist upon transcription of the *csgD* open reading frame and/or of the *csgDEFG* operon needs to be investigated in future studies. 

## 4. Concluding Remarks

Rdar and alternative modes of biofilm formation are delicately regulated. The switch between planktonic and sessile (biofilm) cells denotes not only a radical change in the single cell versus population behavior, but also in the physiology and metabolism of the organism. In *S.* Typhimurium, the activity of the CsgD regulator is pivotal in the stimulation of rdar biofilm formation including modulation of the synthesis of two biofilm promoting factors, curli and cellulose. The synthesis of those biofilm matrix components requires a relevant number of the assets available for the cells. Different environmental stresses including nutrient starvation trigger biofilm formation in *S*. Typhimurium. 

In this report, we describe the Gre factors as new players in the array of regulators affecting *csgD* expression. A potential role for the Gre factors in bacterial adaptation to different environmental stresses has been proposed [3,7]. Hence, expression of *greA* and *greB* occurs under different environmental conditions in *E. coli* [3]. The expression of the *greA* gene is σE-dependent in *E. coli* and *S.* Typhimurium, is induced under hypoxia in *Mycobacterium avium* and under acid stress in *Streptococcus mutans* [34,35,36,37,38]. Mutation in genes encoding Gre factors cause sensitivity to different stresses such as heat shock, osmotic shock and presence of detergents [39,40,41]. A role in stress response for the Gre factor proteins by altering transcriptional expression was suggested for the stress resistant bacterium *Deinococcus radiodurans* [42]. 

In a previous report, we described in *S.* Typhimurium that the Gre factors are required for the expression of *hilD* encoding the main regulator of the SPI-1, a genomic island encoding the type three secretion system 1 needed for invasion of epithelial cells. The the 3′-UTR of the *hilD* gene was identified as the target of the Gre factors [7]. Here, we propose that Gre-mediated alleviation of pausing during 5′-UTR transcription may be pivotal to generate a fully functional *csgD* transcript through nucleolytic cleavage. Such mRNA processing might even influence the effect of additional known post-transcriptional regulators that target the 5′-UTR of *csgD* and can constitute an additional regulatory mechanism in the multilayered regulation of the *csgD* gene. Transcriptional pausing causes gene repression that can be alleviated by the Gre factors as a response to specific environmental stimuli. 

Long 5′UTR are regulatory motifs often involved in post-transcriptional regulation by very diverse mechanisms and targets for regulation at the level of transcription elongation [43]. The RNA chaperone Hfq and small RNAs are among the global regulators that directly target the 5′-UTR of the *csgD* operon to negatively or positively affect transcription [15,16,18]. It has been shown that genes encoding for regulators are especially prone to suffer from transcriptional pausing [3], therefore transcriptional pausing consequently affects the expression of all genes under the control of a specific regulator. Further studies will be required to fully dissect the role of the *csgD* 5′-UTR and the impact of the Gre-dependent pausing on the regulatory pathways that convey on the 5′-UTR of *csgD*.

## Figures and Tables

**Figure 1 microorganisms-10-01921-f001:**
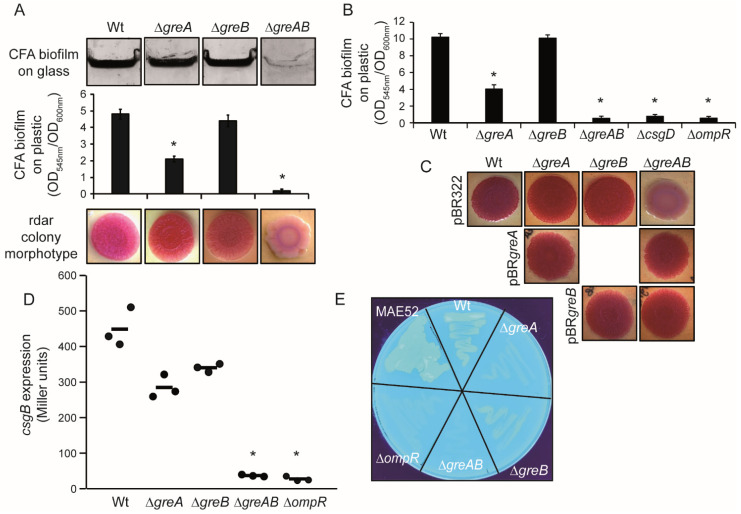
The Gre factors are required for expression of rdar biofilm promoting factors in *S.* Typhimurium. (**A**). Effect of deletion mutations of Gre factor encoding genes on the biofilm formed by the SV5015 strain. Upper panel, crystal violet stained biofilm formed by SV5015 (Wt) and the indicated mutant derivatives strains on glass slides after static growth for 72 h at 25 °C in CFA medium in Hellendahl jars. Middle panel, quantification of crystal violet stained biofilm formed on polystyrene walls of 96 well plates in CFA medium after 72 h at 25 °C. Lower panel, colony morphotype of the indicated strains on Congo red plates. (**B**). Effect of deletion mutations of the Gre factor encoding genes on biofilm formation by the UMR1 strain formed on polystyrene walls of 96 well plates. Cultures and quantification were performed as in (**A**). Biofilm quantification, in both (**A**,**B**), is based on calculation of the mean and standard deviations of six samples (three independent cultures with two technical replicas each). (**C**). Trans-complementation of the UMR1∆*greAB* mutant strain monitored by colony morphotype on Congo red plates carrying either the pBR322 (control vector), pBRgreA or pBRgreB. (**D**). Transcriptional expression of a *csgB::lacZ* chromosomal fusion was monitored in the indicated strains by β-galactosidase determination. Mean and standard deviations calculated from three independent cultures are shown. (**E**). The production of cellulose is impaired in the ∆*greAB* mutant strain. Observed fluorescence under UV of the indicated strains grown at 25 °C for 48 h on a LB_0_ plate supplemented with calcofluor. Statistically significant difference compared to Wt is shown by an asterisk (*p* < 0.0005).

**Figure 2 microorganisms-10-01921-f002:**
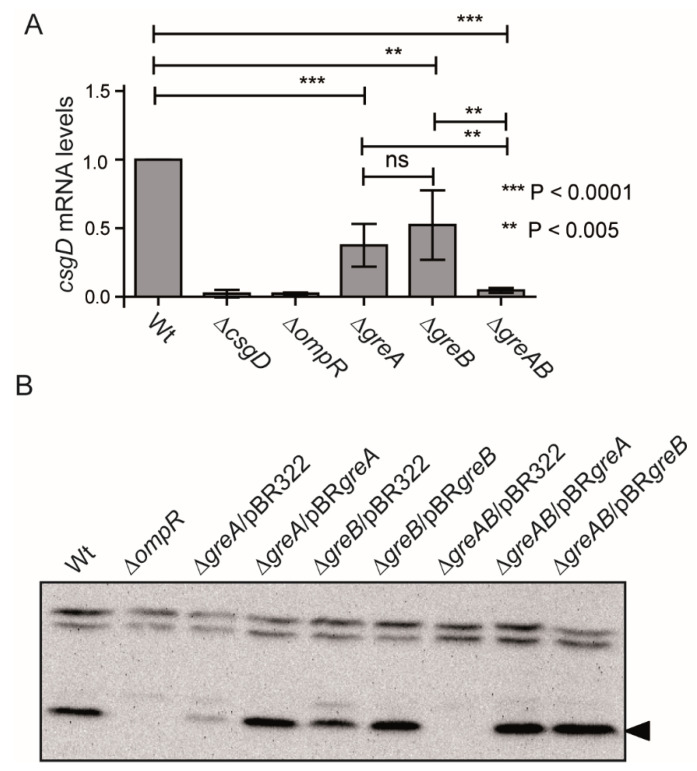
The expression of *csgD* encoding the master regulator of rdar biofilm formation is impaired in the absence of the Gre factors. (**A**). *csgD* mRNA levels were determined by qPCR in RNA samples from cultures of the indicated strains grown on LB_0_ plates at 28 °C for 24 h. Results are normalized to expression of *recA* as an endogenous control and given as relative values to Wt, arbitrarily set as 1.0. The results are mean and standard deviation from three biological replicates. Statistical significance is indicated by *** *p* < 0.0001, ** *p* < 0.005, ns: non-significant. (**B**). Immunodetection of CsgD (indicated by an arrowhead). Cell extracts from cells grown on LB_0_ plates at 28 °C for 24 h were separated by PAGE and analyzed by Western blot analysis.

**Figure 3 microorganisms-10-01921-f003:**
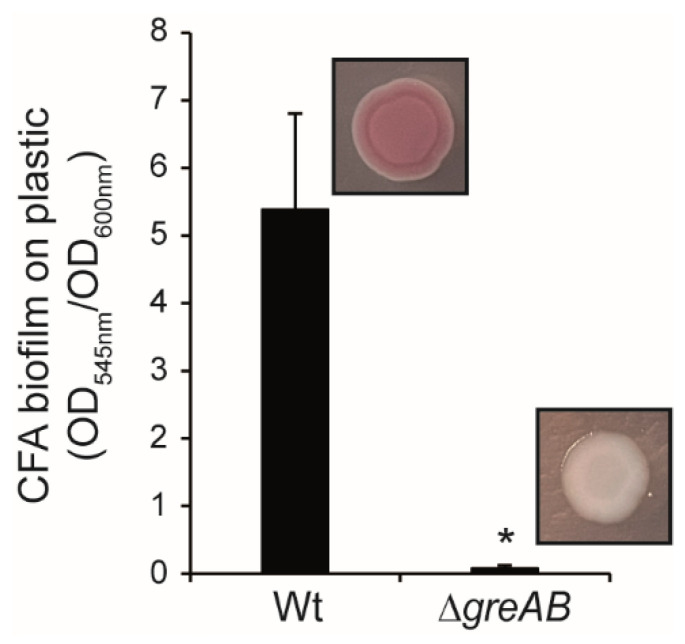
The Gre factors are required for expression of rdar biofilm promotion in *Escherichia coli*. A. Effect of deletion mutants of Gre factor encoding genes on biofilm formation by *E. coli* MG1655 strain. Quantification of crystal violet stained biofilm formed on polystyrene walls of 96 well plates by MG1655 (Wt) and its ∆*greAB* mutant grown in CFA medium for 72 h at 25 °C. Insets show the colony morphotype of the indicated strains on Congo red plates. Biofilm quantification values are mean and standard deviations of six samples. Statistical significance to Wt is shown by an asterisk (*p* < 0.0005).

**Figure 4 microorganisms-10-01921-f004:**
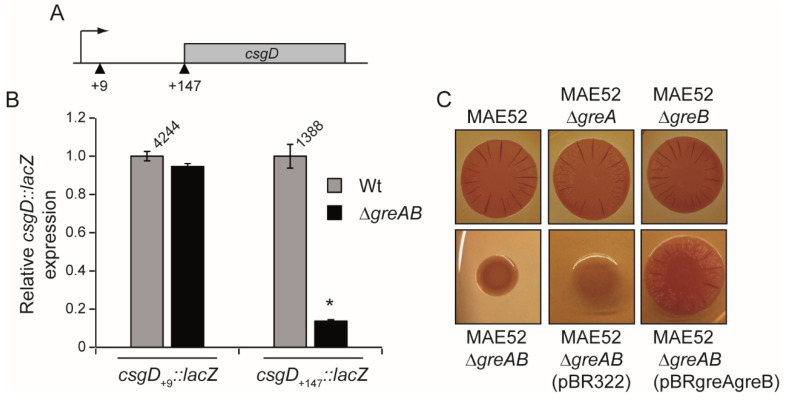
The 5′UTR of the *csgD* gene is the target of the Gre factors. (**A**). Schematic representation of the generated chromosomal *csgD::lacZ* fusion: the *csgD*_+9_*::lacZ* lacks the 5′-UTR and *csgD*_+147_*::lacZ* carries the 5′-UTR. Arrowheads indicate the relative position of the gene fusions. (**B**). Relative expression of *csgD* in Wt and ∆*greAB* in the absence (*csgD*_+9_*::lacZ*) and presence (*csgD*_+147_*::lacZ*) of the 5′UTR. Cultures were grown on LB_0_ plates and incubated at 28 °C for 24 h. A bar shows the arithmetic mean of experimental results and the error bar indicates the standard deviation from three biological replicates. The mean in Miller unit is indicated for the Wt. Statistical significance to Wt is shown by an asterisk (*p* < 0.0005). (**C**). Colony morphotype of strain MAE52 and its derivatives on Congo red agar plate. Plates were incubated at 28 °C for 24 h.

**Figure 5 microorganisms-10-01921-f005:**
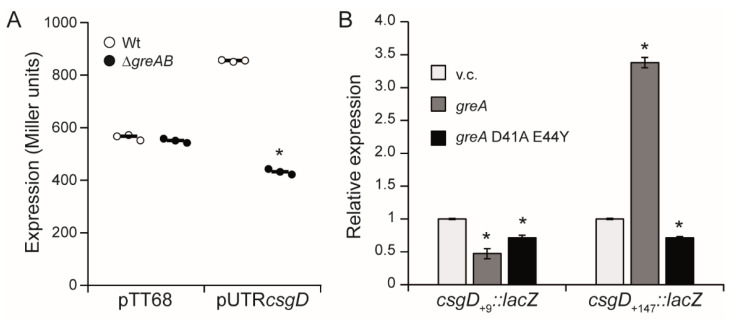
A Gre dependent transcriptional pause is located in the 5′-UTR of the *csgD* gene. (**A**). *lacZ* expression from plasmid construct (pUTR*csgD*) carrying the 5′-UTR of *csgD* cloned in pTT68 vector downstream of the PBAD promoter and upstream of a promoter less *lacZ* gene in both Wt and ∆*greAB* strains. Cultures were grown in LB supplemented with ampicillin and arabinose at 37 °C up to an OD_600nm_ of 2.0. (**B**). Expression from the chromosomal fusions *csgD*_+9_*::lacZ* (lacking the 5′-UTR) and *csgD*_+147_*::lacZ* (carrying the 5′-UTR) in the ∆*greAB* strain carrying the plasmids pHM1883 (vector control, v.c.), pHM1873 (pHM1883 + *greA*_MG1655_) and pHM1854 (pHM1883 + *greA*_MG1655(D41AE44Y)_. Cultures were grown on LB_0_ plates supplemented with spectinomycin and incubated at 28 °C for 48 h. A bar shows the arithmetic mean of experimental results and the error bar indicate the standard deviation from three biological replicates. Statistical significance to Wt is shown by an asterisk (*p* < 0.0005).

## Data Availability

Not applicable.

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
