# Peer review of "Gre Factors Are Required for Biofilm Formation in Salmonella enterica Serovar Typhimurium by Targeting Transcription of the csgD Gene"

_microorganisms, 2022, doi:10.3390/microorganisms10101921_

Round 1
Reviewer 1 Report
Review of Gaviria-Cantin et al. Microorganisms-1866021
The authors describe a series of experiments showing that factors GreA and GreB influence the transcription of CsgD, the master regulator of biofilm formation in Salmonella and E. coli. The impact of GreA/B acts on the 5’ UTR of the csgD mRNA transcript to reduce any transcriptional pausing that might occur. GreA/B have also been shown to regulate hilD expression in SPI-1 type three secretion.
My comments are mostly about how data were presented or explained. I have no major problems with the experiments.
1) Figure 4 – It wasn’t clear to me how the constructs exist – does the +9 strain/construct have the transposon inserted closer to the transcriptional start site in the chromosome? And the +147 construct is inserted further away? The authors need to explain this better
2) Even though the data in Figure 5 is clear, I still have apprehension about the +9 vs +147 construct. Reducing the length of the untranslated region with CsgD impairs the binding of other factors, such as H-NS, which can also affect regulation.
3) Line 48 – Salmonella enterica serovar Typhimurium cannot be shortened to just “Salmonella”, you can say Salmonella ser. Typhimurium, or many researchers write it as S. Typhimurium – follow the convention described by “Antigenic formulae of the salmonella serovars, 2007 - 9th edition; Patrick A.D. Grimont & François-Xavier Weill
4) Lines 253-255 – The way that this data is described is confusing. The results in Figure 3 show that deletion of GreA/B results in loss of biofilm formation
5) Lines 332-333 – The authors are talking about percent identity between E. coli and Salmonella GreA and the presence of residues 41 and 44. This data should be shown as supplementary
6) Figures 1D, 5A – the individual data points should be shown, rather than hidden by the histogram bars – how many measurements were taken?
7) Line 387 – the authors should expand on why this discovery is “remarkable”
Author Response
Responses to Reviewer 1 comments
The authors describe a series of experiments showing that factors GreA and GreB influence the transcription of CsgD, the master regulator of biofilm formation in Salmonella and E. coli. The impact of GreA/B acts on the 5’ UTR of the csgD mRNA transcript to reduce any transcriptional pausing that might occur. GreA/B have also been shown to regulate hilD expression in SPI-1 type three secretion.
My comments are mostly about how data were presented or explained. I have no major problems with the experiments.
- Figure 4 – It wasn’t clear to me how the constructs exist – does the +9 strain/construct have the transposon inserted closer to the transcriptional start site in the chromosome? And the +147 construct is inserted further away? The authors need to explain this better
Response: The description of the experimental procedure for constructing chromosomal lacZ fusions has been substantially revised in the Genetic manipulations section in Materials and methods and missing information on plasmids has been introduced into Table S1. A supplementary material file (File S1), provides further explanation and a schematic diagram of the procedure) In the result section of the amended version it is indicated to the reader (see Materials and methods for details).
Chromosomal lacZ fusions were generated in two subsequent steps. First, homologous recombination by one-step gene replacement (PMID 10829079) was applied to partially delete the csgD gene from different positions (+9 and +147). Subsequently, a promoter less lacZ gene was integrated by site-specific recombination via a suicide plasmid [24] (see File S1 in Supplementary material). Briefly, hybrid primers containing nucleotide sequences homologous to the 5’ (csgD+9 and csgD+147) and 3’-end (csgD lac Rev) and sequences flanking the antibiotic marker were used to amplify the kanamycin resistance cassette encoded by plasmid pKD4. Gene replacement and removal of the antibiotic cassette in S. Typhimurium SV5015 was performed by the standard approach (PMID 10829079). Constructs were PCR-verified with primers csgD up/csgD down located in the sequences flanking the deleted open reading frame (Table S2). The suicide plasmid pKG136, bearing a FRT site upstream of a promoter less lacZ gene, was integrated into the chromosome by FLP-mediated recombination aided by plasmid pCP20 (PMID 12062810). Recombinant clones were selected on LB agar containing kanamycin and X-Gal. Primers used in this work are listed in Table S2 (Supplementary material).
- Even though the data in Figure 5 is clear, I still have apprehension about the +9 vs +147 construct. Reducing the length of the untranslated region with CsgD impairs the binding of other factors, such as H-NS, which can also affect regulation.
Response: The reviewer is correct indicating that the different gene fusions, by having different length of the UTR may affect regulation by other regulatory factors. Indeed, we know that small RNAs are binding in the 5’-UTR equally as there is a IHF binding site in the csgD open reading frame. All our data here, however, indicate that the effect of the Gre factors seems to be direct by alleviating a transcriptional pause located between +9 and +147. However, we cannot rule out that the effect could be indirect by affecting either the expression of an unknown factor or the binding of this unknown factor that targets the +9 to +147 sequence. We can also not rule out that an effect observed within the 5’-UTR cannot be modulated by downstream sequences. A new paragraph has been introduced in the Results and Discussion section of the amended version of the manuscript:
Although, in vitro transcription experiments will be required to confirm the direct involvement of the Gre factors in csgD transcription, our data suggest that GreA and GreB alleviate transcriptional arrest in the 5’-UTR of csgD mediated by the residues participating in transcript cleavage by the RNA polymerase. Whether additional GreA and GreB checkpoints exist upon transcription of the csgD open reading frame and/or of the csgDEFG operon needs to be investigated in future studies.
3) Line 48 – Salmonella enterica serovar Typhimurium cannot be shortened to just “Salmonella”, you can say Salmonella ser. Typhimurium, or many researchers write it as S. Typhimurium – follow the convention described by “Antigenic formulae of the salmonella serovars, 2007 - 9th edition; Patrick A.D. Grimont & François-Xavier Weill
Response: Changes have been done as suggested.
4) Lines 253-255 – The way that this data is described is confusing. The results in Figure 3 show that deletion of GreA/B results in loss of biofilm formation
Response: The text has been amended in the revised version of the manuscript to describe in detail the results in Figure 3 as follows: Not unexpectedly, the absence of the Gre factors causes a severe drop in rdar biofilm production and in the expression of biofilm promoting factors, as depicted by the rdar morphotypes, also in E. coli (Fig. 3). In summary, greA and greB upregulate the central hub of the rdar biofilm, csgD, on the steady state mRNA level in S. Typhimurium and presumably also in E. coli.
5) Lines 332-333 – The authors are talking about percent identity between E. coli and Salmonella GreA and the presence of residues 41 and 44. This data should be shown as supplementary.
Response: A sequence alignment of the GreA proteins from MG1655 (E. coli) and ATCC14028/SL1344 (S. Typhimurium strains) is now shown as Figure S1 in the Supplementary material.
6) Figures 1D, 5A – the individual data points should be shown, rather than hidden by the histogram bars – how many measurements were taken?
Response: The figures 1D and 5A are now shown as suggested by the reviewer.
7) Line 387 – the authors should expand on why this discovery is “remarkable”
Response: The sentence has been modified in the amended version of the manuscript, as follows: It has been shown that genes encoding for regulators are especially prone to suffer of transcriptional pausing [3], therefore the consequences of transcriptional pausing have an effect in the expression of all genes under the control of the specific regulator.

Reviewer 2 Report
This study is reported the important rule of Gre factors in Salmonella enterica serovar Typhimurium when the bacterium forms biofilms. This study takes great information that will be highly contributed to combat Salmonella related infectious diseases by regulating biofilm formation (inhibiting Curli expression and cellulose production.
I think this manuscript should be encouraged for future readers to understand more clearly. I would like you to mention some comments about following things.
1) Please indicate information of suppliers about experimental materials: the name, City, Country.
2) Bacterial names are described in Italic style.
3) In the Results, some sentences (as below) are discussion, thereby, the authors should reconsider to separate Results and Discussion, or combine them and make a conclusion part.
L.250-252: Altogether, our data indicated that the Gre factors are required for the expression of the master regulator of biofilm formation in Salmonella CsgD. Gre factors as well as the curli operon are conserved between S. typhimurium and Escherichia coli.
L.277-278: Thus we can speculate that, as in most cases, transcriptional pauses that occur in 5’untranslated regions (UTR) [3] are relieved by the Gre factors in the case of transcription at the csgD promoter.
L.294-301: These results clearly indicate that the 5’-UTR is required for the Gre-mediated stim- 293 ulation of csgD expression and that the regulation does not occur at the level of transcription initiation. The latest statement was further corroborated using the strain MAE52 that carries a single point mutation in the csgD promoter causing an approx. 3-fold higher transcriptional expression of the csgD gene. In MAE52, csgD is transcribed with the aid of the 297 housekeeping sigma factor RpoD instead of the stress sigma factor RpoS as in strain UMR1 [11,31]. Independently that the csgD gene was expressed from a semi-constitutive promoter in MAE52, mutations in greA and greB lead to a downregulation of the expression of the rdar morphotype further supporting a mechanism of action downstream of transcription initiation (Fig. 4C).
L.311-316: These data indicate that the 5’-UTR of the csgD gene is the target of the Gre factors. The csgD 5’-UTR was cloned in the pTT68 plasmid, downstream of a pBAD promoter and upstream of a promoter less lacZ gene. Expression of the construct was induced by the addition of arabinose and monitored in both Wt and DgreAB genetic backgrounds. The expression dropped in the DgreAB mutant with the construct carrying the csgD 5’-UTR316 (Fig. 5A) further indicating that the 5’-UTR is the target of the Gre factors.
L.329-346: Gre factors promotes transcription elongation by resolving backtracking of paused transcription complexes [1]. In E. coli, the D41 and E44 GreA residues are essential for the ability of GreA to alleviate backtracked complexes and thereby to suppress transcriptional 331 pauses [32]. E. coli and Salmonella GreA share high percentage of identity (96.8 %) comprising the 41 and 44 residues. Previously, we demonstrated that E. coli GreA complements the DgreAB mutant in Salmonella and that the D41 and E44 residues are required to suppress transcriptional pausing [7]. To determine whether the downregulation of csgD expression in the Gre-factor deficient strain is due to a transcriptional pause, two GreA 336 variants of E. coli were used, the GreAWt and the GreAD41AE44Y, being the latter impaired to resolve paused transcription complexes. The effect of the presence of GreAWt and GreAD41AE44Y on the expression of the csgD gene was monitored using the csgD+9::lacZ and csgD+147::lacZ constructs (Fig. 5B). As expected attending to the previous results, the expression of GreAWt in a DgreAB mutant complemented the absence of the Gre factors when 341 using the csgD fusion carrying the 5’-UTR (csgD+147::lacZ), whereas does not induce expression in the csgD fusion lacking the 5’-UTR (csgD+9::lacZ). Remarkably, no induction of the 343 csgD+147::lacZ construct was detected when GreAD41AE44Y was used, indicating that the reported effect of the Gre factors on csgD expression can be due to the resolution of a transcriptional pause occurring in the 5’-UTR of the csgD gene.
Author Response
Responses to Reviewer 2 comments
This study is reported the important rule of Gre factors in Salmonella enterica serovar Typhimurium when the bacterium forms biofilms. This study takes great information that will be highly contributed to combat Salmonella related infectious diseases by regulating biofilm formation (inhibiting Curli expression and cellulose production.
I think this manuscript should be encouraged for future readers to understand more clearly. I would like you to mention some comments about following things.
- Please indicate information of suppliers about experimental materials: the name, City, Country.
Response: The information of suppliers is now indicated in the amended version of the manuscript
2) Bacterial names are described in Italic style.
Response: We apologize for the mistakes in the submitted version of the manuscript. All the bacterial names are now in italics.
3) In the Results, some sentences (as below) are discussion, thereby, the authors should reconsider to separate Results and Discussion, or combine them and make a conclusion part.
Response: As indicated by the reviewer, in the amended version of the manuscript the Results and Discussion sections have been combined and a Concluding remarks section has been included.
L.250-252: Altogether, our data indicated that the Gre factors are required for the expression of the master regulator of biofilm formation in Salmonella CsgD. Gre factors as well as the curli operon are conserved between S. typhimurium and Escherichia coli.
L.277-278: Thus we can speculate that, as in most cases, transcriptional pauses that occur in 5’untranslated regions (UTR) [3] are relieved by the Gre factors in the case of transcription at the csgD promoter.
L.294-301: These results clearly indicate that the 5’-UTR is required for the Gre-mediated stim- 293 ulation of csgD expression and that the regulation does not occur at the level of transcription initiation. The latest statement was further corroborated using the strain MAE52 that carries a single point mutation in the csgD promoter causing an approx. 3-fold higher transcriptional expression of the csgD gene. In MAE52, csgD is transcribed with the aid of the 297 housekeeping sigma factor RpoD instead of the stress sigma factor RpoS as in strain UMR1 [11,31]. Independently that the csgD gene was expressed from a semi-constitutive promoter in MAE52, mutations in greA and greB lead to a downregulation of the expression of the rdar morphotype further supporting a mechanism of action downstream of transcription initiation (Fig. 4C).
L.311-316: These data indicate that the 5’-UTR of the csgD gene is the target of the Gre factors. The csgD 5’-UTR was cloned in the pTT68 plasmid, downstream of a pBAD promoter and upstream of a promoter less lacZ gene. Expression of the construct was induced by the addition of arabinose and monitored in both Wt and DgreAB genetic backgrounds. The expression dropped in the DgreAB mutant with the construct carrying the csgD 5’-UTR316 (Fig. 5A) further indicating that the 5’-UTR is the target of the Gre factors.
L.329-346: Gre factors promotes transcription elongation by resolving backtracking of paused transcription complexes [1]. In E. coli, the D41 and E44 GreA residues are essential for the ability of GreA to alleviate backtracked complexes and thereby to suppress transcriptional 331 pauses [32]. E. coli and Salmonella GreA share high percentage of identity (96.8 %) comprising the 41 and 44 residues. Previously, we demonstrated that E. coli GreA complements the DgreAB mutant in Salmonella and that the D41 and E44 residues are required to suppress transcriptional pausing [7]. To determine whether the downregulation of csgD expression in the Gre-factor deficient strain is due to a transcriptional pause, two GreA 336 variants of E. coli were used, the GreAWt and the GreAD41AE44Y, being the latter impaired to resolve paused transcription complexes. The effect of the presence of GreAWt and GreAD41AE44Y on the expression of the csgD gene was monitored using the csgD+9::lacZ and csgD+147::lacZ constructs (Fig. 5B). As expected attending to the previous results, the expression of GreAWt in a DgreAB mutant complemented the absence of the Gre factors when 341 using the csgD fusion carrying the 5’-UTR (csgD+147::lacZ), whereas does not induce expression in the csgD fusion lacking the 5’-UTR (csgD+9::lacZ). Remarkably, no induction of the 343 csgD+147::lacZ construct was detected when GreAD41AE44Y was used, indicating that the reported effect of the Gre factors on csgD expression can be due to the resolution of a transcriptional pause occurring in the 5’-UTR of the csgD gene.
